# Electrophilic Agonists Modulate the Transient Receptor Potential Ankyrin-1 Channels Mediated by Insulin and Glucagon-like Peptide-1 Secretion for Glucose Homeostasis

**DOI:** 10.3390/ph16081167

**Published:** 2023-08-16

**Authors:** Marisa Jadna Silva Frederico, Andreza Cipriani, Jocelyn Brice Alexandre Heim, Ana Karla Bittencourt Mendes, Marcela Aragón, Joana Margarida Gaspar, Nylane Maria Nunes De Alencar, Fátima Regina Mena Barreto Silva

**Affiliations:** 1Laboratory of Hormones & Signal Transduction, Departament of Biochemistry, Center of Biological Sciences, Campus Trindade, Federal University of Santa Catarina, Florianópolis 88040-900, SC, Brazil; andrezacipriani@hotmail.com (A.C.); jocelynheim@hotmail.com (J.B.A.H.); akmendes29@hotmail.com (A.K.B.M.); joana.gaspar@ufsc.br (J.M.G.); 2Laboratory of Biochemistry and Pharmacology, Departament of Pharmacology and Physiology, Drug Research and Development Center (DRDC), Medical School, Federal University of Ceará, Rua Coronel Nunes de Melo, Fortaleza 60430-275, CE, Brazil; nylane@ufc.br; 3Departament of Pharmacy, Science School, National University of Colombia, Bogotá 11011, Colombia; dmaragonn@unal.edu.co

**Keywords:** nutrients, intestinal hormone, beta-cells, calcium, glycemia, dissacharidase, ionic channels

## Abstract

This pre-clinical study investigated the transient receptor potential ankyrin-1 (TRPA1) channels on modulating targets for glucose homeostasis using agonists: the electrophilic agonists, cinnamaldehyde (CIN) and allyl isothiocyanate (AITC), and the non-electrophilic agonist, carvacrol (CRV). A glucose tolerance test was performed on rats. CIN and AITC (5, 10 and 20 mg/kg) or CRV (25, 100, 300, and 600 mg/kg) were administered intraperitoneally (i.p.), and glycemia was measured. In the intestine, Glucagon-like peptide-1 (GLP-1) and disaccharidase activity were evaluated (in vivo and in vitro, respectively). Furthermore, in vivo and in vitro insulin secretion was determined. Islets were used to measure insulin secretion and calcium influx. CIN and AITC improved glucose tolerance and increased insulin secretion in vivo and in vitro. CRV was unable to reduce glycemia. Electrophilic agonists, CIN and AITC, inhibited disaccharidases and acted as secretagogues in the intestine by inducing GLP-1 release in vivo and in vitro and contributed to insulin secretion and glycemia. The effect of CIN on calcium influx in pancreatic islets (insulin secretion) involves voltage-dependent calcium channels and calcium from stores. TRPA1 triggers calcium influx and potentiates intracellular calcium release to induce insulin secretion, suggesting that electrophilic agonists mediate this signaling transduction for the control of glycemia.

## 1. Introduction

Epidemiological studies point out high fat and high carbohydrate diets, associated with low physical activity, as the important risk factors for the development of type 2 diabetes mellitus. This problem had been aggravated by the isolation caused by the coronavirus disease pandemic (COVID-19), which changed the routine to a more sedentary lifestyle [1].

Glucose homeostasis is mainly regulated by insulin and glucagon. After a meal that contains carbohydrates, beta-cells release insulin. Insulin, through its signaling pathway, will decrease glycemia until normal levels are achieved [2]. Under type 2 diabetes, the presence of insulin resistance does not occur, and hyperglycemia remains high after a carbohydrate meal.

Experimental evidence indicates that transient receptor potential (TRP) channels contribute to glucose homeostasis [3]. The TRP is subdivided into seven subfamilies: TRPC (canonical), TRPV (vanilloid), TRPM (melastatin), TRPP (polycystin), TRPML (mucolipin), TRPA (ankyrin), and TRPN (NOMPC-like). TRPM2, TRPM4, TRPA1, and TRPV1 channels are involved in the secretion of insulin in β-cells [3,4,5], making them a promising target for diabetes treatment. However, upstream and downstream activation of TRPA1 for signal transduction and insulin secretion are far from understood.

Compounds from plants used in condiments and foods activate transient potential receptor A1 (TRPA1); these include allylisothiocyanate (mustard, wasabi, cabbage, and horseradish), cinnamaldehyde (cinnamon), allicin (garlic), and carvacrol (oregano) (see chemical structure in Figure 1) [6]. TRPA1 channels are permeable to sodium (Na^+^) and calcium (Ca^2+^) and have a single conductance channel. The TRPA1 receptor can be activated in two ways: first by electrophilic agonists that form covalent ducts with cysteine and lysine residues contained within the N-terminal domain in the cytosolic portion of the channel, causing its opening [7]. Secondly, by non-electrophilic ligands, via traditional orthosteric ligand/receptor binding [8]. Cinnamaldehyde, allylisothiocyanate, and gallic acid are electrophilic ligands, while carvacrol is a non-electrophilic ligand. On the one hand, studies show that TRPA1 ligands are reactive electrophilic agents that bind critically to cysteine residues on the ankyrin portion of the channel [8]. On the other hand, a wide range of non-electrophilic compounds that activate TRPA1 are proposed, including dihydropyridines [9], monoterpenes [10], alcohols [11], and alkaloids [12]. The binding preferences of these non-electrophilic compounds and the structure–function studies based on a chimeric approach using the non-electrophilic agonist menthol revealed crucial residues in the transmembrane segment 5, which are essential for the activation of TRPA1 [13]. Thus, although a wide variety of chemicals have been identified as electrophilic and non-electrophilic inducers of TRPA1 bioactivity, the mechanism that mediates its activation in glycemia remains elusive. In this study, the activation of TRPA1 channels was investigated, as well as their involvement in the regulation of glycemia and in vivo and in vitro incretin and insulin secretion. In addition, the acute effects of TRPA1 agonists on intestinal disaccharidase activity were investigated using pharmacological modulation by cinnamaldehyde, allylisothiocyanate, and carvacrol. Additionally, calcium influx in pancreatic islets was used to investigate the downstream pathways triggered by the activation of TRPA1 and its involvement in insulin secretion.

## 2. Results

### 2.1. Role of Transient Receptor Potential Ankyrin-1 in Maintaining Glucose Tolerance

The basic chemical structures of the three TRPA1 agonists selected are shown below: Figure 1A, cinnamaldehyde; Figure 1B, allylisothiocyanate; Figure 1C, carvacrol. To assess whether TRPA1 is involved in glucose homeostasis, three different agonists of TRPA1 were selected: CIN (5, 10, and 20 mg/kg), AITC (5, 10, and 20 mg/kg), and CRV (25, 100, 300, and 600 mg/kg). Each compound was administered 30 min before glucose (4 g/kg). CIN (20 mg/kg; via i.p.) significantly reduced serum glucose by approximately 18% and 32% at 30 and 60 min, respectively. Surprisingly, 10 mg/kg CIN significantly increased glycemia at 30 and 60 min. No effect on serum glucose lowering was detected for 5 mg/kg when compared with the hyperglycemic control group (Figure 2A).

Treatment with AITC (5 mg/kg) reduced serum glucose by approximately 28% at 60 min, with AITC (10 mg/kg) decreased serum glucose by about 21% and 35% after 30 and 60 min, respectively. The highest dose of AITC tested (20 mg/kg) significantly reduced serum glucose by 16% and 22% after 30 and 60 min, respectively (Figure 2B). On the other hand, the non-electrophilic TRPA1 agonist, CRV, had no effect on glycemia, even at higher doses (300 and 600 mg/kg) (Figure 2C).

### 2.2. Influence of Transient Receptor Potential Ankyrin-1 Agonists on the Activity of Intestinal Disaccharides

Other mechanisms by which TRPA1 agonists modulate glucose homeostasis were also investigated. The effects of the TRPA1 agonists, CIN, AITC, and CRV (1, 10 and 100 μM) on maltase, sucrase, and lactase activity were evaluated (Figure 3A–C). CIN (100 μM) inhibited the activity of maltase by approximately 50%. This compound also reduced sucrase activity by approximately 15% at the three concentrations used and inhibited lactase activity by about 80% when used at a concentration of 100 μM. AITC inhibited the activity of maltase by approximately 40%, 75%, and 75% at 1, 10, and 100 μM, respectively. AITC at 1, 10, and 100 μM inhibited sucrase activity by approximately 85%, 80%, and 40%, respectively. In addition, AITC reduced the activity of lactase by about 31% at all concentrations tested. CRV has no effect on the activity of any of the disaccharidases studied.

### 2.3. Effects of Transient Receptor Potential Ankyrin-1 Agonists on Glucagon-like Peptide-1 Secretion

To evaluate whether the antihyperglycemic actions of CIN and AITC are mediated by increased Glucagon-like peptide-1 (GLP-1) secretion, this incretin was measured by collecting serum during the glucose tolerance test, after administration of CIN and AITC, in vivo. The glucose tolerance test augmented GLP-1 secretion [14]. GLP-1 was significantly increased in animals treated with CIN by approximately 20% and 29% at 30 and 60 min, while AITC increased GLP-1 by about 44% and 60% at 15 and 60 min, respectively, when compared with the respective hyperglycemic control group (Figure 4A,B). In addition, the in vitro incubation of the intestinal colon with CIN increased GLP-1 secretion by around 280% in relation to the control group. Glucose (11 mM) was used as a positive control for this experiment (Figure 4C).

Table 1 shows that CIN presented an incretinogenic index that was 1.34 times greater than that of the hyperglycemic control, confirming the increase in GLP-1 secretion in vivo (Figure 4A) and in vitro (Figure 4C). AITC presented an incretinogenic index of 1.7 times greater than that of the hyperglycemic control, demonstrating its high GLP-1 secretory power when compared to CIN.

### 2.4. Role of Transient Receptor Potential Ankyrin-1 on In Vivo and In Vitro Insulin Secretion

The most effective dose of CIN (20 mg/kg) for serum glucose lowering was initially used as the base for insulin secretion after in vivo treatment. For the CIN group, insulin secretion increased by approximately 100% at 30 and 60 min, respectively, when compared to the control hyperglycemic group (Figure 5A). For the in vitro experiments, 100 μM CIN increased static insulin secretion by approximately 314% in isolated pancreatic islets (Figure 5B). The immunofluorescence images of the pancreatic islet sections of rats show the nucleus (DAPI), β-cells (insulin), and merged cells (Figure 5C).

Similarly, AITC had a similar effect on in vivo insulin during the glucose tolerance test and in vitro static insulin secretion from pancreatic islets, constituting a powerful secretagogue effect. After AITC treatment in vivo, insulin secretion increased by 87% at 60 min compared to the hyperglycemic control group (Figure 6A). In vitro, AITC increased insulin secretion by 230% compared with the control group (Figure 6B). The insulinogenic index is commonly used to assess early phase insulin secretion in response to glucose and quantify the secretion of insulin relative to β-cell function (Table 2) [15].

CIN presented an insulinogenic index that was 2.13-fold higher than that of the hyperglycemic control group. AITC treatment demonstrated an insulinogenic index of 2.47-fold higher than that of the hyperglycemic control. These results confirm the data presented in Figure 5A and Figure 6A and are consistent with the findings of the in vitro experiments (Figure 5B and Figure 6B, respectively).

### 2.5. Mechanism by Which Transient Receptor Potential Ankyrin-1 Agonists Modulate Calcium Influx Signaling Pathways during Insulin Secretion

The lipophilic CIN agonist altered glucose homeostasis and presented greater potential as an approach for diabetes therapy when compared with AITC and CRV. Thus, CIN was chosen for the investigation of its mechanism of action on calcium influx in isolated pancreatic islets. To study the involvement of K^+^-ATP channels in the stimulatory effect of CIN on calcium influx in pancreatic islets, glibenclamide, a K^+^-ATP channel blocker, and diazoxide, a K^+^-ATP channel activator, were used. CIN significantly increased the calcium influx in pancreatic islets when compared to the control. The presence of diazoxide or glibenclamide had no effect on the modulation of calcium influx by CIN, demonstrating that CIN does not act directly on K^+^-ATP channels to trigger calcium influx (Figure 7A). These data are in agreement with those of Ashcroft and Rorsman [16], who reported that calcium influx can also be produced or enhanced by mechanisms that are independent of the plasma membrane depolarization induced by the K^+^-ATP channel activity.

Subsequently, the roles of stored calcium and the voltage-dependent calcium channels (VDCC) in the stimulatory effect of CIN on calcium influx were evaluated. BAPTA-AM (an intracellular calcium chelator) and nifedipine (a VDCC blocker) were used. The stimulatory effect of CIN on calcium influx was inhibited by approximately 27% by BAPTA-AM, indicating the participation of intracellular calcium in its mechanism of action. Accordingly, nifedipine reduced the calcium influx caused by CIN by about 26%, indicating that VDCC-mediated external calcium influx partly mediates the mechanism of action of this compound (Figure 7B).

We then investigated the effects of the TRPV1 channel agonist, capsaicin [17], on the calcium influx stimulated by CIN. Capsaicin, both alone and together with CIN, increased the calcium influx in pancreatic islets in a similar manner to that of CIN. On the other hand, in the presence of a TRPA1 channel inhibitor, HC-030031, CIN-induced calcium influx was completely abolished (Figure 7C). In addition, amiloride, a sodium channel blocker, had no effect on CIN-induced calcium influx, indicating that sodium ions do not participate in this mechanism in pancreatic islets (Figure 7D). Figure 8 shows the detailed mechanism of action.

## 3. Discussion

There is a great need for further development of new effective and safer antidiabetic drugs for the management of glycemia. A molecular target of particular interest is TRPA1, which is expressed in β-cells but not in alpha and delta cells. TRPA1 can detect hyperglycemia and transmit several stimuli involved in insulin secretion. The oral glucose tolerance test performed in this study has a sensitivity of approximately 15% for detecting a statistically significant impact of test compounds. In this study, both CIN and AITC reduced glycemia in a non-dose-dependent manner. Other authors have reported similar results, showing that oral administration of cinnamic acid, but not CIN, improved glucose tolerance in a dose-dependent manner in streptozotocin-induced diabetic rats [18]. As reported in the literature, CIN (20 mg/kg) attenuated hyperglycemia in the fatty-sucrose diet/streptozotocin rat model of gestational diabetes. The addition of 0.1%, 0.5%, and 1% of *Cinnamomum tamala* oil to the diet reduced glycemia in mice with diabetes induced by a high fat and high sucrose diet, and CIN was found to represent the major component of this oil [19]. The controversial beneficial dose–response effects of CIN on the metabolism of glucose are in agreement with those discussed in the literature. For example, in humans, results obtained following cinnamon consumption (where 70% possess CIN) are highly controversial. Some studies reported no alteration in glycemia [20,21,22], while others showed that cinnamon consumption could be an excellent adjunct in the treatment of diabetes [23]. In addition, the effect of AITC on glucose tolerance occurred in accordance with the previous data reported in the literature. Sahin et al. [24] showed that AITC reduced serum glucose levels in type 2 diabetic rats induced by a high-fat diet/streptozotocin.

CRV, which is a non-electrophilic agonist of TRPA1, reduced blood glucose only when associated with rosiglitazone in type 2 diabetic C57Bl/6J mice induced by a high-fat diet [25]. The action of CRV on hyperglycemia, detected with our approach, may be due to its action on adrenaline, glucagon, or corticosterone secretion, which may directly affect the enzymes involved in glycogenolysis and/or the gluconeogenesis pathway [26,27,28]. Taking our data into account, it can be concluded that the site of activation of the TRPA1 receptor at the cysteine and lysine residues was directly related to its antihyperglycemic activity. These data indicate that the activation of the TRPA1 receptor is mediated by electrophilic agonists.

Maltase and sucrase are the enzymes that are most studied as therapeutic targets for diabetes [29]. Clinical studies in humans have shown that patients with diabetes mellitus have high disaccharidase enzyme activity in the small intestine [30] and propose that this is the main cause of the increase in postprandial glucose levels [31]. As far as we know, this is the first report of the activation of TRPA1 receptors in disaccharidase enzyme activity. Our data indicate that the agonists, CIN and AITC, exert their activities by inhibiting these enzymes, whereas the non-electrophilic CRV agonist did not alter the activities of the disaccharidases. However, the intestine should be further explored as a target for the electrophilic agonists of TRPA1 that regulate the disaccharidases.

TRPA1 channels have been detected by immunohistochemistry on the epithelial surface of the rat colon and the murine small intestine and colon [32,33]. CIN and AITC were found to increase GLP-1 secretion in vivo. Kim and colleagues showed that methyl seringate, a TRPA1 agonist, suppressed food consumption and gastric emptying but did not increase GLP-1 levels in mice [34]. In another study, TRPA1 activation caused the increased secretion of GLP-1 in primary mouse gut and GLUTag cells [35]. These results suggest the stimulation of GLP-1 secretion downstream from the ligand-gated ion channel, TRPA1.

CIN and AITC treatments increased insulin secretion in vivo and in vitro. Accordingly, cinnamic acid stimulated in vitro insulin secretion [18]. The binding of agonists to the TRPA1 receptor in pancreatic β-cells was associated with insulin secretion. Furthermore, depolarization evoked by TRPA1 binding was blocked by inhibiting the K^+^-ATP channel [3]. Notably, glibenclamide, a second-generation sulfonylurea, blocks the K^+^-ATP channel as its main mechanism of action but also activates TRPA1 channels, and this was related in part to its secretagogue capacity [35]. Thus, we attribute part of the antihyperglycemic effects of TRPA1 agonists, observed in the glucose tolerance test, to their ability to induce insulin secretion. Therefore, the mechanism by which CIN induces calcium influx in isolated pancreatic islets was evaluated.

The mechanism of K^+^-ATP channel closure is known to cause cell depolarization and trigger the insulin exocytosis process [11]. In this study, K^+^-ATP channels did not participate in CIN-induced calcium influx. Image and patch clamp techniques demonstrated that the activation of TRPA1 by exogenous and endogenous agonists causes insulin secretion through calcium influx and cell depolarization [3]. The influence of an internal calcium chelator and the calcium channel blocker on the stimulatory effect of CIN on calcium influx supports, for the first time, that the depolarization caused by the entry of calcium through the TRPA1 channel is mediated by the activation of L-type voltage-dependent calcium channels, thereby releasing intracellular calcium and culminating in insulin secretion.

TRPV1 and TRPA1 channels are functionally expressed in pancreatic β-cells, INS-1 cells and RINm5F cells [36]. Furthermore, TRPA1 has a 20% amino acid sequence homology with TRPV1, where the activation of this channel is also related to increased insulin secretion [37]. Capsaicin, a TRPV1 agonist [16], known for its insulin secretagogue action on pancreatic β-cells, did not alter the effect of CIN on calcium influx [38]. On the other hand, HC-030031, a selective TRPA1 channel blocker [3] that antagonizes AITC and formalin-evoked calcium influx, completely blocked the stimulatory effect of CIN. As such, the distinct participation of TRPA1 in calcium influx was highlighted.

In the terminal nerves, the activation of TRPA1 results in membrane depolarization, mainly due to Na^+^ influx and induces a local calcium influx and further neuropeptide exocytosis via Ca^2+^-dependent pathways [39]. Similarly to the mechanism described in the terminal nerves, we evaluated whether the influx of calcium, induced by CIN, could be stimulated by Na^+^ influx for further cellular depolarization and induction of insulin secretion. However, when we evaluated the influence of Na^+^ ions on the influx of calcium in isolated pancreatic islets induced by CIN, no change was detected. These data are in line with those observed using tetrodotoxin (TTX), a voltage-dependent sodium channel blocker, which did not cause any alteration in insulin secretion in the RIN strain [3].

Electrophilic TRPA1 agonists are incretin- and insulinogenic compounds that contribute to glucose homeostasis. Based on the antihyperglycemic activities of these edible agonists of plant origin and their popular use as antidiabetics, we reported for the first time their interaction with incretins for insulin secretion and for the control of glycemia. In addition, electrophilic TRPA1 agonists inhibit the activity of disaccharidases, indicating the intestine as a target for the electrophilic activation of TRPA1. Furthermore, pharmacological findings suggest an electrophilic component of TRPA1-mediated effects on strategic targets of glucose homeostasis.

## 4. Materials and Methods

### 4.1. Animals

Male Wistar rats (190–220 g) were monitored and maintained in accordance with the ethical recommendations of the local Committee for Ethics in Animal Research of the UFSC (Protocol CEUA/UFSC/PP00398/749) and the Guide for the Care and Use of Laboratory Animals. The animals were maintained with pelleted food (Nuvital, Nuvilab CR1, Curitiba, PR, Brazil), while tap water was available ad libitum.

### 4.2. Effects of Cinnamaldehyde, Allylisothiocyanate and Carvacrol on Glucose Tolerance, Serum Glucagon-like Peptide-1, and Insulin Secretion

Fasted rats (12 h) were divided into groups of 5 rats for the glucose tolerance test: hyperglycemic control (received 4 g glucose/kg of body weight plus vehicle, i.p.); hyperglycemic plus cinnamaldehyde (CIN) (5, 10, and 20 mg/kg, i.p.) or allylisothiocyanate (AITC) (5, 10, and 20 mg/kg, i.p.) or carvacrol (CRV) (25, 100, 300, and 600 mg/kg, i.p.). Glycaemia was measured before any treatment or glucose overload (zero time represents fasting glycemia). Rats received CIN, AITC, and CRV, and glucose (4 g/kg of body weight) was administered 30 min later. Glycemia was measured at 15, 30, 60, and 180 min after glucose overload by the glucose oxidase method. Serum insulin and Glucagon-like peptide-1 (GLP-1) were measured by ELISA using 4 rats in each group. For GLP-1 in vivo measurements, sitagliptin (3 mg/kg, i.p.) was administered 1 h before treatments in all groups to prevent the degradation of GLP-1 [40]. Doses of CIN, AITC, and CRV were administered based on previous studies [18,41].

### 4.3. Effects of Cinnamaldehyde, Allylisothiocyanate and Carvacrol on Intestinal Disaccharidase Activity (In Vitro)

The duodenum was removed, washed in 0.9% NaCl solution, cut into small pieces (2.5 cm each), incubated with 2 mL of saline, and treated with CIN, AITC, or CRV (1, 10, and 100 μM) for 20 min. After the incubation, tissues were homogenized, and an aliquot was saved for total protein measurements. The homogenates were incubated for 10 min in 100 μL of maleate buffer (pH 6.0) with respective substrates: maltose, sucrose, and lactose in a concentration of 0.11 M. Maltase (EC 3.2.1.20), lactase (EC 3.2.1.23), and sucrase (EC 3.2.1.48) activities were determined by the glucose method oxidase. The specific activity was defined as the activity of the enzyme (U) per mg protein (corresponding to the amount of enzyme that catalyzes a reaction with a rate of formation of 1 micromol of product *per* minute). The total protein concentration was determined [42].

### 4.4. Isolation of Intestinal Colon Slices for Static Glucagon-like Peptide-1 Measurement

The intestinal colon was removed, washed with saline at 2 °C, and dissected into slices. Slices in Krebs Ringer-bicarbonate buffer (KRb-buffer) containing 5 mM glucose, at 37 °C, pH 7.4, were then preincubated for 60 min in a metabolic incubator and carbon dioxide atmosphere with O_2_:CO_2_ (95:5, *v*/*v*). Afterwards, the slices were incubated for 10 min in KRb-buffer and without (control) or with 100 μM CIN. The supernatant was collected, and aliquots were used to quantify GLP-1.

### 4.5. Insulin and Glucagon-like Peptide-1 Measurements

Detection of insulin and GLP-1 was performed by enzyme-linked immunosorbent assay (ELISA) (Millipore, Research Park Drive, MO, USA), used for the quantitative determination of rat serum and static insulin (catalog no. EZRMI-13K), and rat serum GLP-1 (catalog no. EZGLP1T-36K) was performed according to the manufacturer’s instructions.

Serum samples were analyzed in duplicate, and the results were expressed as ng/dL of insulin and pM/dL of GLP-1. Static insulin (in KRb-buffer) and GLP-1 secretion mL^−1^ were measured and expressed as ng of insulin/μg of protein (15 islets per well) and pM GLP-1/dL (slice of incubation). The incremental areas under the response curves (AUC) were calculated. The insulinogenic and incretinogenic indices were calculated as the ratios between AUC_insulin_ and AUC_glucose_ (from zero to 60 min) and/or AUC_GLP-1_ and AUC_glucose_ (from zero to 60 min) and expressed as ng of serum insulin/mg of plasma glucose and pM of serum GLP-1/mg of plasma glucose, respectively [43].

### 4.6. Immunofluorescence Images of the Islet Architecture for β-Cell Staining

The pancreas of rats was removed and incubated in KRb HEPES buffer for 1 h (95% O_2_/CO_2_: 5%, *v*/*v*) at 37 °C. After incubation, the tissues were kept inside cassettes, fixed in 4% paraformaldehyde, and previously solubilized with phosphate buffer (pH 7.3). The tissues were dehydrated using a gradient with increased concentrations of ethanol. Samples were included in paraffin, sliced into sections (5 mm) [44]**,** and arranged in glass slides. The slides were dewaxed in xylol and rehydrated in an ethanol gradient. As such, the tissues were blocked with PBS—3% BSA for 60 min at 4 °C and incubated for 12 h at 4 °C with polyclonal anti-insulin sc-9168 (1: 200) antibody obtained from Santa Cruz Biotechnology (Dallas, TX, USA) diluted in PBS—3% BSA. Afterwards, the tissues were incubated with secondary IgG anti-rabbit antibody conjugated to Alexa Fluor 647, purchased from Cell Signaling Technology, Inc. (Danvers, MA, USA), diluted (1:1000) in PBS—3% BSA, for 60 min at room temperature. Subsequently, samples were washed with PBS and incubated with 40,6-diamidino-2-phenylindol (DAPI) purchased from Invitrogen, Thermo Fisher Scientific (Carlsbad, CA, USA) [45] for 10 min in the dark and at room temperature. The tissues adhered to a coverslip with FluorSave reagent purchased from Calbiochem Merkmilipore (Darmstadt, Germany). Control staining was performed on serial slides where the primary anti-insulin antibody was replaced by PBS—3% BSA. The images were captured in an Axio Scan slide scanner with a Zeiss Blue image capture system (Plan-Apochromat 20×/0.8 M27 lens; excitation and emission filters red: 653—668 nm, blue: 353–465 nm). For each rat, at least 10 images in TIF format of the pancreatic islets (40× magnification) were analyzed with the Image J 1.40 program (Waynw Rasband, National Institutes of Health, USA). All lighting and rinsing conditions were kept constant.

### 4.7. Isolation of Pancreatic Islets for Studies on Calcium Influx and Static Insulin Measurement

The rat pancreas was visualized through a central abdominal incision. The bile duct was clamped at the tip of the duodenum and cannulated at a point sufficiently proximal to the liver. Krebs Ringer-bicarbonate buffer (KRb) with a composition of 122 mM NaCl, 3 mM KCl, 1.2 mM MgSO_4_, 1.3 mM CaCl_2_, 0.4 mM KH_2_PO_4_, and 25 mM NaHCO_3_, supplemented with 8 mM HEPES (KRb-HEPES) medium, was introduced slowly into the bile duct using a syringe until the pancreas was clearly distended. The pancreas was then removed and maintained in Petri dishes with KRb-HEPES medium. The pancreatic tissue was cut into small pieces (2 × 2 mm) and incubated in plastic tubes in 1 mL of KRb-HEPES medium supplemented with collagenase (3 mg). After incubation, the mixture was transferred to a tube (110 × 15 mm), resuspended in 10 mL with collagenase-free medium, and centrifuged at room temperature for 3 min at 45× *g* in an Excelsa Baby centrifuge (model 206), FANEM, São Paulo, SP, Brazil. The supernatant was discarded, and the sediment was resuspended in KRb-HEPES medium. This washing procedure was repeated five times; for the last two washes, the islets were allowed to settle without centrifugation. Thus, aliquots (100 µL) of the final sediment with the isolated islets were transferred to tubes with the KRb-HEPES incubation medium [46]. This improved technique has become the gold standard for rodent islet isolation [47].

### 4.8. Study of the Effects of Transient Receptor Potential Ankyrin-1 Agonists/Antagonists on the Stimulatory Effect of Cinnamaldehyde on Calcium Influx in Pancreatic Islets

Isolated pancreatic islets were preincubated in KRb-HEPES containing 5 mM glucose, 0.1 μCi/mL ^45^Ca^2+^ for 60 min in a metabolic incubator at 37 °C, and pH 7.4 in a carbon dioxide atmosphere with O_2_:CO_2_ (95: 5, *v*/*v*). The islets were incubated in KRb-HEPES, without (control) or with 100 μM CIN [48]. In some experiments, the channel or receptor antagonists/agonists were added after 45 min of incubation, corresponding to 15 min prior to the treatment and maintained throughout the incubation period. The following compounds were used: 250 µM diazoxide; 20 nM glibenclamide; 50 µM BAPTA-AM; 1 µM nifedipine [49]; 100 µM capsaicin [50]; 100 µM HC-030031 [3]**;** 3 µM amiloride [51]. Islets were treated for 10 min, and the incubation was stopped with-ice cold lanthanum buffer [52]. Aliquots of the samples were placed in liquid scintillation vials in an LKB rack, on a beta liquid scintillation spectrometer (LS 6500 model; Multi-Purpose Scintillation Counter-Beckman Coulter, Boston, MA, USA), for the measurements of radioactivity. Total protein quantification was determined [42].

### 4.9. Statistical Analyzes

Data are expressed as means ± S.E.M. One-way analysis of variance (ANOVA), followed by the Bonferroni post hoc test, or unpaired Student’s *t*-test, was used to determine the significance of differences between groups. Differences were considered to be significant at *p* ≤ 0.05.

## 5. Conclusions

Electrophilic TRPA1 agonists are incretin- and insulinogenic compounds that contribute to glucose homeostasis. Based on the antihyperglycemic activities of these edible agonists of plant origin and their popular use as antidiabetics, we reported for the first time their interaction with incretins for insulin secretion and for the control of glycemia. In addition, electrophilic TRPA1 agonists inhibit the activity of disaccharidases, indicating the intestine as a target for the electrophilic activation of TRPA1. Furthermore, pharmacological findings suggest an electrophilic component of TRPA1-mediated effects on strategic targets of glucose homeostasis.

## Figures and Tables

**Figure 1 pharmaceuticals-16-01167-f001:**
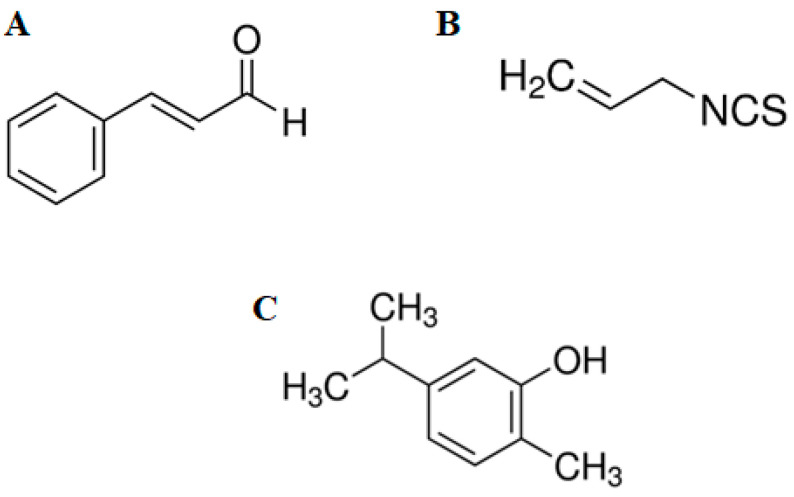
Chemical structures of cinnamaldehyde (CIN) (**A**), allylisothiocyanate (AITC) (**B**), and carvacrol (CRV) (**C**).

**Figure 2 pharmaceuticals-16-01167-f002:**
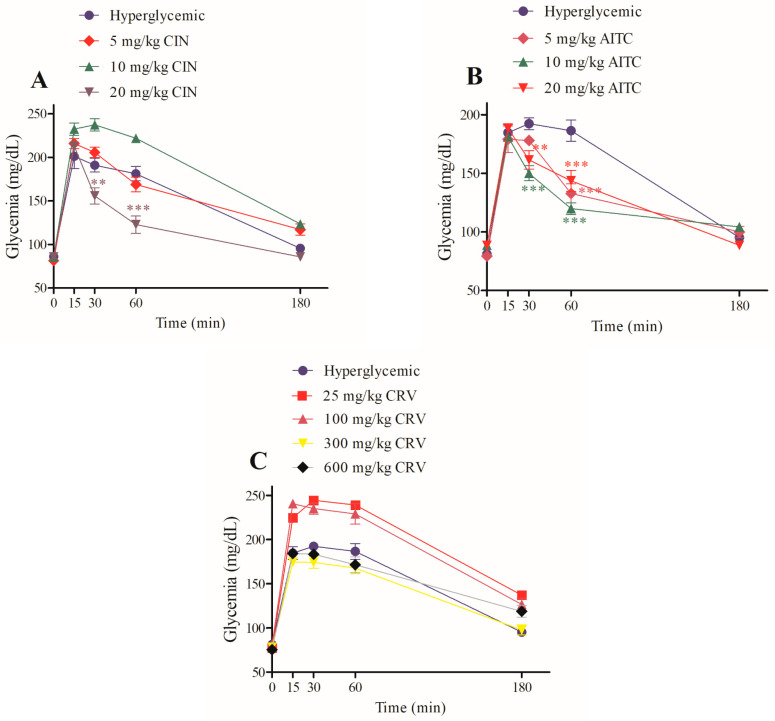
Effect of CIN (5, 10, and 20 mg/ kg) (**A**), AITC (5, 10, and 20 mg/ kg) (**B**), and CRV (25, 100, 300, and 600 mg/ kg) (**C**) on glucose tolerance curves in intraperitoneally treated rats. Values are expressed as mean ± SEM; *n* = 6. ** *p* ≤ 0.01 and *** *p* ≤ 0.001; compared to the hyperglycemic control group or control group.

**Figure 3 pharmaceuticals-16-01167-f003:**
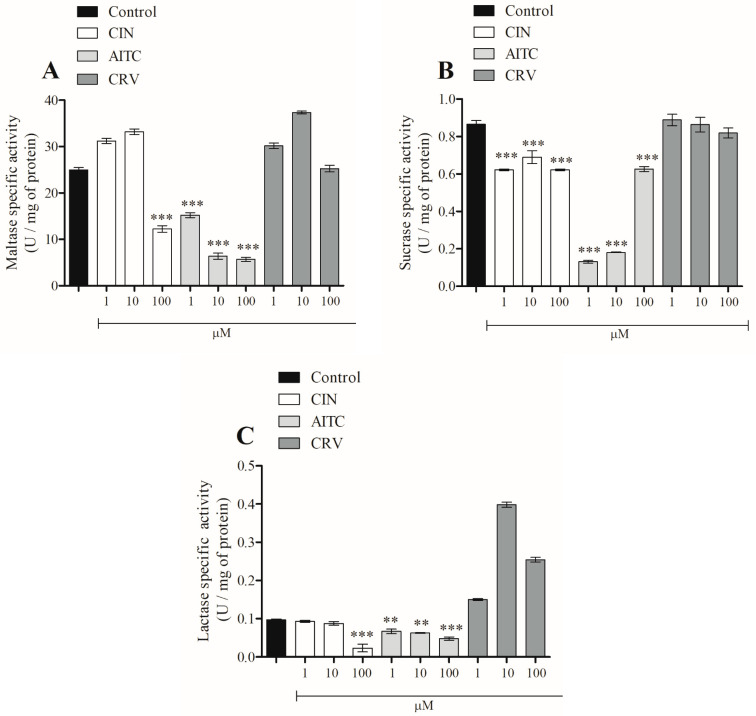
In vivo effects of CIN, AITC, and CRV on intestinal disaccharidase activities; maltase (**A**), sucrose (**B**), and lactase (**C**). Pre-incubation time with treatments: 20 min. Incubation time with substrates: 10 min. Values are expressed as means ± SEM; *n* = 6. ** *p* ≤ 0.01 and *** *p* ≤ 0.001; when compared to the control group.

**Figure 4 pharmaceuticals-16-01167-f004:**
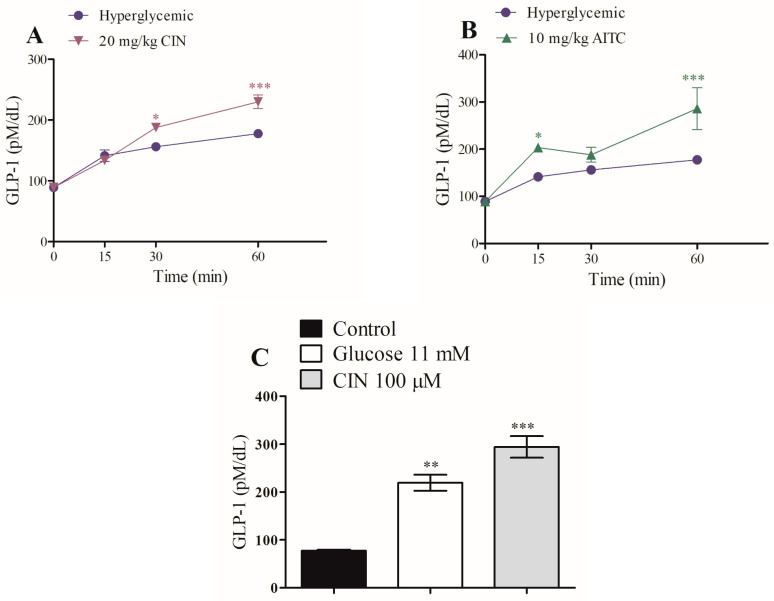
In vivo effect of intraperitoneal CIN (20 mg/kg) (**A**) and AITC (10 mg/kg) (**B**) on serum GLP-1 levels in hyperglycemic rats. Values are expressed as means ± SEM; *n* = 6. In vitro effect of (**C**) CIN (100 μM) on static GLP-1 secretion in intestinal colon slices, *n* = 5. * *p* ≤ 0.05, ** *p* ≤ 0.01, and *** *p* ≤ 0.001 compared to the hyperglycemic control group or control group.

**Figure 5 pharmaceuticals-16-01167-f005:**
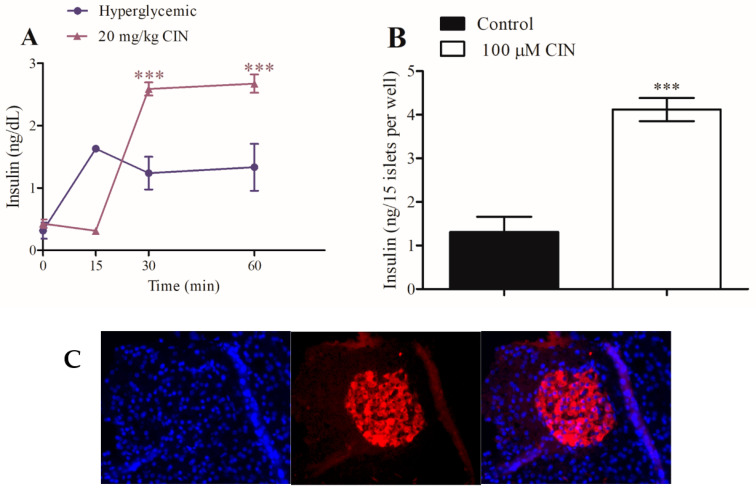
Effect of CIN on serum insulin secretion in hyperglycemic intraperitoneally-treated rats (**A**) and static insulin secretion (**B**) from isolated pancreatic islets. Pancreatic islet sections of rats show the DAPI (control group), insulin group (β-cells), and merged cells (**C**). Values are expressed as means ± SEM; *n* = 6. *** *p* ≤ 0.001 compared to the hyperglycemic control group or control group.

**Figure 6 pharmaceuticals-16-01167-f006:**
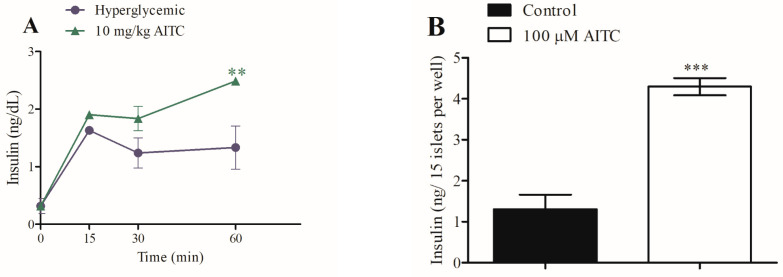
Effect of intraperitoneally administered AITC on serum insulin in hyperglycemic rats (**A**); *n* = 5 and on static insulin secretion from isolated pancreatic islets (**B**). Values are expressed as means ± SEM; *n* = 6. ** *p* ≤ 0.01 and *** *p* ≤ 0.001 compared to the hyperglycemic control group or control group.

**Figure 7 pharmaceuticals-16-01167-f007:**
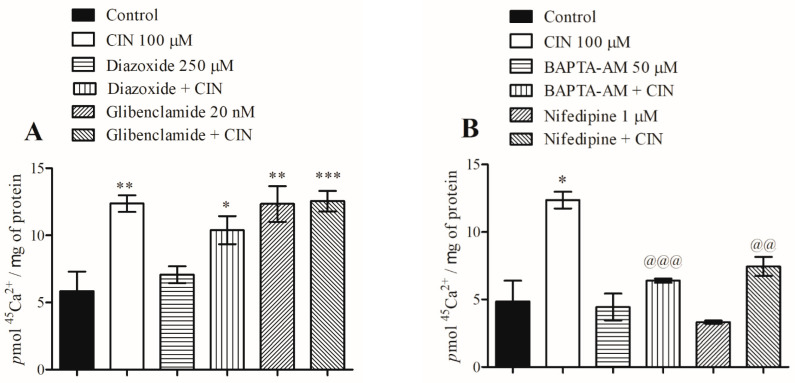
Effect of CIN on calcium influx in pancreatic islets (**A**–**D**). Islets were pre-incubated with blockers and agonists for 15 min and during the incubation. Pre-incubation = 60 min; incubation = 10 min. Values are expressed as means ± SEM; *n* = 6. * *p* ≤ 0.05, ** *p* ≤ 0.01, and *** *p* ≤ 0.001 compared to the control group. ^@@^ *p* ≤ 0.01 and ^@@@^ *p* ≤ 0.001 compared to the CIN group.

**Figure 8 pharmaceuticals-16-01167-f008:**
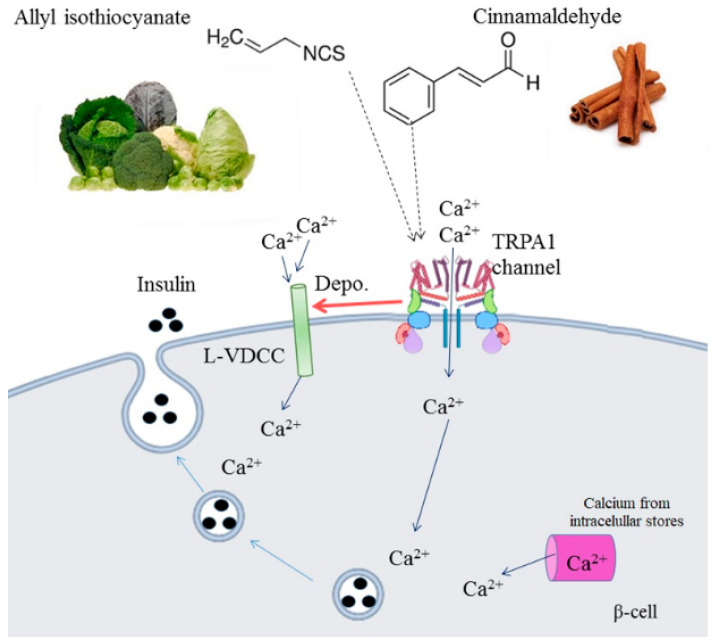
Schematic representation of the proposed mechanism of electrophilic agonists of TRPA1 channels, CIN (100 μM), and AITC (100 μM) on calcium influx and insulin secretion in pancreatic islets. Ca^2+^, calcium ion; L-VDCC, Type L voltage-dependent calcium channels; TRPA1, transient receptor potential ankyrin-1.

**Table 1 pharmaceuticals-16-01167-t001:** Incretinogenic index of TRPA1 agonists.

Group	Incretinogenic Index (pM/mg)
Hyperglycemic (4 g/kg, i.p.)	0.8393
CIN (20 mg/kg, i.p.)	1.1258 ***
AITC (10 mg/kg, i.p.)	1.4306 ***

Incretinogenic index (Inc. I.). *** *p* ≤ 0.001 compared to the control hyperglycemic group.

**Table 2 pharmaceuticals-16-01167-t002:** Insulinogenic Index for TRPA1 agonists.

Group	Insulinogenic Index (ng/mg)
Hyperglycemic (4 g/kg, i.p.)	0.4805
CIN (20 mg/kg, i.p.)	1.0250 ***
AITC (10 mg/kg, i.p.)	1.1888 ***

Insulinogenic Index (Ins. I.). *** *p* ≤ 0.001 compared to the control hyperglycemic group.

## Data Availability

Data is contained within the article.

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
