# Peer review of "Electrophilic Agonists Modulate the Transient Receptor Potential Ankyrin-1 Channels Mediated by Insulin and Glucagon-like Peptide-1 Secretion for Glucose Homeostasis"

_pharmaceuticals, 2023, doi:10.3390/ph16081167_

Round 1

Reviewer 1 Report

The incidence of diabetes is significant and the identification of new therapeutic solutions is appreciated. The proposed outline is interesting, but the presentation of the graphics is disappointing. It is necessary to redo the graphs with an improved resolution (possibly using free programs even if these do not implicitly pass the statistical significance on the graph). The differences between the in-vtro and in-vivo effects may be due to pharmacokinetic elements, but also to some elements avoided by the authors, such as those in pharmacodynamics. Reaching the therapeutic targets of the active compounds can induce differences in-vivo compared to in-vitro. The authors mentioned sedentarism in their analysis, but it would have been useful to address its possible effects on the microvascular structure, for example. A more detailed opinion of the authors regarding the differences between the two experimental models is necessary (at least one - two sentences).

Author Response

Reviewer 1 Response

The authors thank to the referee suggestions.

Comments and Suggestions for Authors

The incidence of diabetes is significant, and the identification of new therapeutic solutions is appreciated. The proposed outline is interesting, but the presentation of the graphics is disappointing. It is necessary to redo the graphs with an improved resolution (possibly using free programs even if these do not implicitly pass the statistical significance on the graph).

Response: We did new graphs with better resolution and improves the comprehension of each one.

The differences between the in-vitro and in-vivo effects may be due to pharmacokinetic elements, but also to some elements avoided by the authors, such as those in pharmacodynamics. Reaching the therapeutic targets of the active compounds can induce differences in-vivo compared to in-vitro. The authors mentioned sedentarism in their analysis, but it would have been useful to address its possible effects on the microvascular structure, for example. A more detailed opinion of the authors regarding the differences between the two experimental models is necessary (at least one - two sentences).

Response: The in vivo and in vitro approach are complementary, in terms of glucose regulation. Additionally, differences may be to several factors. In fact, we did not study the aspects of pharmacokinetic or pharmacodynamics and so it is difficult to discuss these possible influences. We believe that it is another two interesting step to explore.

Sincerely

Fátima RMB Silva

Reviewer 2 Report

The manuscript is interesting. The authors use an up-to-date methodology. The results support the discussion. However, I believe it is necessary to make some changes to the manuscript.

I. Major comments:

1. In the introduction it is necessary to include a brief paragraph on metabolic and dietary aspects related to glucose homeostasis, especially in the context of obesity - insulin resistance and finally type 2 diabetes.

2. It would be interesting for the authors to discuss the role of oxidative stress in the development of insulin resistance, particularly the impact on the receptor.

3. Glucose homeostasis is dependent on carbohydrate intake. It is necessary to discuss this point.

II. Minor comments:

1. Improve the wording of the study objective

2. Improve the resolution of figure 2

3. Is figure 1 necessary? It is not cited in the text, or I could not find it cited.

4. Improve the resolution of figures 3, 4, 5, 6 and 7. It is difficult to identify the experimental groups

The manuscript is well written, but some editorial errors need to be corrected.

Author Response

Reviewer 2

Comments and Suggestions for Authors

The manuscript is interesting. The authors use an up-to-date methodology. The results support the discussion. However, I believe it is necessary to make some changes to the manuscript.

  1. Major comments:
  2. In the introduction it is necessary to include a brief paragraph on metabolic and dietary aspects related to glucose homeostasis, especially in the context of obesity - insulin resistance and finally type 2 diabetes.

Response: It was revised and a reference as included.

  1. It would be interesting for the authors to discuss the role of oxidative stress in the development of insulin resistance, particularly the impact on the receptor.

Response: The authors understand that oxidative stress was not addressed in this study. Therefore, a discussion of this would lead to confusion.

  1. Glucose homeostasis is dependent on carbohydrate intake. It is necessary to discuss this point.

Response: We did not understand what the referee meant, exactly.

  1. Minor comments:
  2. Improve the wording of the study objective

Response: It was revised.

  1. Improve the resolution of figure 2

Response: The figure was improved.

  1. Is figure 1 necessary? It is not cited in the text, or I could not find it cited.

Response: Thank. It was revised.

  1. Improve the resolution of figures 3, 4, 5, 6 and 7. It is difficult to identify the experimental groups

Response: All figures were revised, and new figures were uploaded.

Comments on the Quality of English Language

The manuscript is well written, but some editorial errors need to be corrected.

Response: It was revised.

Sincerely

Fátima RMB Silva

Round 2

Reviewer 2 Report

Authors answered all my comments. Therefore, the manuscript can be accepted in the present form.